# Unconventional Myosins: How Regulation Meets Function

**DOI:** 10.3390/ijms21010067

**Published:** 2019-12-20

**Authors:** Natalia Fili, Christopher P. Toseland

**Affiliations:** Department of Oncology and Metabolism, Medical School, University of Sheffield, Sheffield S10 2RX, UK

**Keywords:** unconventional myosins, regulation, dimerization, auto-inhibition, cargo recognition, cargo transporters, anchors, alternative splicing, binding partners, load, local environment

## Abstract

Unconventional myosins are multi-potent molecular motors that are assigned important roles in fundamental cellular processes. Depending on their mechano-enzymatic properties and structural features, myosins fulfil their roles by acting as cargo transporters along the actin cytoskeleton, molecular anchors or tension sensors. In order to perform such a wide range of roles and modes of action, myosins need to be under tight regulation in time and space. This is achieved at multiple levels through diverse regulatory mechanisms: the alternative splicing of various isoforms, the interaction with their binding partners, their phosphorylation, their applied load and the composition of their local environment, such as ions and lipids. This review summarizes our current knowledge of how unconventional myosins are regulated, how these regulatory mechanisms can adapt to the specific features of a myosin and how they can converge with each other in order to ensure the required tight control of their function.

## 1. Introduction

Myosins are molecular engines that utilise the energy released from ATP hydrolysis to produce mechanical work along the actin cytoskeleton. Together with the microtubule-based dynein and kinesins, myosins are part of the large family of cytoskeletal motors. In humans, the myosin superfamily consists of 40 myosin genes, which are classified into 12 classes, on the basis of their architecture [1,2]. These are further divided into two groups: the group of conventional myosins which include the skeletal, cardiac and smooth muscle myosins and the non-muscle myosin II; and the group of unconventional myosins which represent two-thirds of myosin genes in humans [3]. Unconventional myosins perform key roles in a broad range of fundamental cellular processes, including endocytosis, exocytosis, intracellular trafficking, organelle and plasma membrane morphology, cell adhesions, cell motility and transcription [4,5,6] (Figure 1). Depending on their mechano-enzymatic properties and structural features, they fulfil these roles by acting as cargo transporters, as molecular anchors onto the actin cytoskeleton, as tension sensors, as actin cross-linkers and even as regulators of the actin cytoskeleton itself [5,7].

In terms of their structure, myosins consist of three main domains: the motor head, the neck region and the tail domain (Figure 2). The motor is an 80 kDa domain which contains the nucleotide binding pocket, where ATP hydrolysis occurs, and the actin binding site. During the ATPase cycle, conformation changes within the motor are translated into movement along actin filaments [8]. Despite the conserved nature of their motor domain, myosins differ in terms of the maximal actin-activated ATPase rate, as well as the time the motor stays tightly bound to actin, known as the duty ratio [6,9]. Following the motor head, myosins contain a neck region of variable lengths, depending on the number of isoleucine–glutamine (IQ) motifs, typically ranging between one to six. The role of the neck region is to act as a level arm, which amplifies the conformational changes that occur in the motor domain during the ATPase cycle. The number of IQ motifs determines the length of the lever arm and as a result the step size of the myosin [10]. These IQ motifs have been shown to interact with a broad range of light chains including calmodulin (CaM), regulatory light chains (RLC) and essential light chains (ELC), with CaM being the prevalent one [11]. Finally, following their neck region, myosins contain a highly diverse myosin-specific tail region. This tail region can contain: a) long coiled-coil regions, for instance in myosin V [12] and myosin X [13], or a short coiled-coil in myosin VI [14,15], which mediate myosin dimerization, b) helical regions that can be used to extend the length of the lever arm, as it has been shown for myosin VI [15], c) single α-helix (SAH) regions that also serve as lever arm extensions, for example in myosin VI [16], VIIa [17], and X [18], d) phospholipid binding domains, like the pleckstrin homology (PH) of myosin I and X, e) various protein-interacting domains which mediate the interaction with their binding partners such as the Src homology 3 (SH3) in myosin I and VII; the MyTH4-FERM domain (myosin tail homology 4–band 4.1, ezrin, radixin, moesin) in myosin VII and X, the PDZ (PSD95/Dlg/ZO1) domain in myosin XVIII and f) domains with enzymatic activity, like the kinase domain in myosin III and the RhoGAP domain in myosin IX [6]. Therefore, it is not a surprise that the structural content of the tail region is the one that defines the oligomeric state of the myosin, its cargo recognition and its intracellular localization.

In addition, the content of the tail also determines the conformation of the myosin. As it will be discussed in detail below, some myosins have been shown to adopt a back-folded auto-inhibitory conformation, with the aim to prevent futile consumption of ATP. The dimeric myosin Va is one of the best examples of a myosin under auto-inhibitory regulation, whereby the dimer adopts a compact, back-folded conformation in which the two globular tail domains (GTD) interact with the two motor domains [19,20]. This conformer exhibits low actin activated ATPase activity at low Ca^2+^ concentrations [21]. This intramolecular interaction not only decreases the ATPase rate, but also significantly reduces the motor’s duty ratio, thus making it dissociate from the actin [22]. Similar auto-inhibited conformations associated with low actin-activated ATPase activity have been also shown for the monomeric myosin VII [23,24] and myosin X [25].

Given their multi-functionality and involvement in key cellular processes, myosins need to be under tight regulation both in time and space. Various mechanisms have evolved in order to ensure that the right myosin is activated at the right time; that it adopts the appropriate conformation for its function; and that it interacts with the suitable binding partner in order to associate with the appropriate cargo and to be recruited at the correct intracellular location. To this end, the regulation of myosins occurs at multiple levels: from gene expression and post-translational modifications, to the dynamic manipulation of their local environment which includes divalent cations, lipids, binding partners and applied load (Figure 2). This review will present an overview of the current knowledge about the mechanisms that the cell deploys in order to regulate the activity and function of unconventional myosins (Table 1). We will discuss how these mechanisms specifically adapt to the features and cellular roles of each myosin and how they can converge in order to achieve the tight regulation required (Table 2). Finally, we will highlight aspects of the regulation that remain elusive and how these can be addressed in the future.

## 2. Regulation by Alternative Spicing

One of the mechanisms regulating unconventional myosins occurs at the transcription level through alternative splicing. Splicing occurs in a cell type or tissue type dependent manner and yields various spliced variants, which can exhibit different properties, intracellular localization and functions.

One myosin which is regulated by alternate splicing is myosin VI. Splicing occurs in two regions within its tail, one N-terminal to its cargo binding domain (CBD) and one within the CBD, resulting in the addition of a 21–31 amino acid (aa) large insert (LI) and/or a 9 aa small insert (SI), respectively [26]. Therefore, myosin VI can be expressed as four spliced variants, depending on the cell and tissue type: the non-insert (NI) isoform, the LI, the SI and the one containing both SI+LI inserts. This alternative splicing has been shown to modulate the intracellular targeting and therefore the function of this multifunctional motor. In polarized epithelia cells, the LI isoform is specifically recruited at the apical microvilli where it plays a role in clathrin-mediated endocytosis [26]. In neurosecretory cells, the SI isoform regulates exocytosis by anchoring the secretory granules to the cortical actin network [27]. As it will be mentioned below, the SI contains a c-Src kinase phosphorylation DYD motif which is required for this function [27], highlighting the interplay between different modes of regulation. The NI isoform, which is widely expressed, is associated with uncoated endocytic vesicles regulating endocytosis [28] and, in polarized cells, mediates the transport of tyrosine motif–containing proteins to the basolateral membrane [29]. Recently, new light has been shed into the mechanism through which isoform splicing regulates myosin VI intracellular targeting (Figure 3). Structural studies by Wollscheid et al. revealed that the LI forms a regulatory helix which, on the one hand masks the RRL binding motif preventing interactions with RRL binding partners, and on the other hand provides a new isoform-specific clathrin-binding site [30]. Interestingly, as it will be discussed in more detail below, the masked RRL binding motif has been shown to be the higher affinity site [31], compared to the second interaction motif WWY, where the clathrin-coated vesicle adaptor Disabled-2 (Dab2) is known to bind [32]. In this way, the presence of LI favours the otherwise low affinity interactions with WWY-binding partners, such as Dab2 and explains the preferential localization of LI isoform on clathrin-coated pits. The impact of this regulatory mechanism to the intracellular localization of myosin VI is also highlighted by the fact the LI isoform is excluded from the nucleus [14]. Instead, the NI isoform, where the RRL site is accessible, is the one recruited to the nucleus, through a mechanism probably dependent on an RRL binding partner. Of note, this isoform, which has been shown to regulate transcription [14], is the one that is overexpressed in aggressive, metastatic cancers [30]. These findings highlight how isoform splicing and binding partner regulation are integrated in order to achieve tight control of this multi-functional myosin.

Alternative splicing within the tail region has also been shown to regulate myosin V. Three exons, namely exons B, D and F, which are located between the coiled-coil region and the GTD of myosin Va, can be alternatively spliced in a tissue-specific manner [33,34,35,36]. Similarly, myosin Vb can contain exons A, B, C, D and E [37]. Alternative splicing of these exons yields various myosin V variants that interact with different adaptor proteins, therefore modulating the association of myosin V with diverse intracellular cargo. For instance, the melanocyte-specific isoform containing the exon F encoded sequence is required for the interaction of myosin Va with melanophilin and its targeting to melanosomes through a tripartite complex with the small GTPase Rab27a [38,39]. The interaction with melanophilin has been shown to release the auto-inhibited back-folded conformation of myosin Va and to stimulate its actin-activated ATPase activity [40]. Similarly, the three-amino acid exon B is required for the association of myosin Va with the cargo adaptor dynein light chain 2 (DLC2) [41,42]. Finally, the aa sequence encoded by exon D is required for the interaction of myosin Va and Vb with the small GTPase Rab10 [37].

Another myosin regulated by alternative splicing is the tension-sensitive myosin Ib. In this case, splicing occurs within the light chain binding domain, which is the region acting as the lever arm [43]. Splicing results in three isoforms containing 4 (myosin Ib(c)), 5 (myosin Ib(b)) or 6 (myosin Ib(a)) non-identical IQ motifs. Although alternative splicing does not affect the ATPase properties of the isoforms, the IQ motifs have differential affinity for CaM and the different isoforms have different motility rates, which depends on the length of lever arm and the affinity of the IQ motifs for CaM [44]. Moreover, isoform splicing has been shown to regulate not only the step size of myosin Ib, but also its force sensing capability [45]. Due to this differential force sensitivity, different isoforms can respond to different mechanical challenges.

Isoform splicing has also been reported for other classes of myosins. In myosin XVIIIa, isoform splicing controls the presence (isoform α) or absence (isoform β) of the N-terminal PDZ-domain, regulating in this way the subcellular localization of the isoforms [46]. In class III myosins, alternative splicing has been reported to regulate the number of IQ motifs, as well as the length of the tail of isoform myosin IIIb, which might be linked to Bardet-Biedl syndrome [47]. Alternate splicing also regulates the presence or absence of a 1203 aa region, N-terminal to the motor domain of myosin XVa, which is thought to play an important role in normal hearing [48]. However, the biological role of these variants and their link to disease have not been addressed yet.

Although alternative splicing emerges an important regulator of myosins’ function, there is still a lot to discover about the underlying mechanistic details of such regulation. This would require a holistic research approach combining *in vitro* biochemical characterization of the spliced variants with *in cellulo* studies to address their physiological impact.

## 3. How the Load Upon a Myosin Modulates Its Function

Force is a regulatory cue that modulates various cellular processes, including cell and organelle morphology, cell motility and intracellular signalling. Therefore, it is not a surprise that force can regulate the mode of action of some unconventional myosins.

The monomeric class I myosins are a characteristic example of force sensitive motors. Single molecule analysis of the widely expressed myosin-Ib revealed that small opposing loads (<2 pN) dramatically increase its attachment lifetime on actin more than 75-fold [49]. In this way, mechanical tension can tune the duty ratio of this myosin I isoform between a low (<0.2, absence of load) and a high (>0.9, presence of load) value. In addition to myosin Ib, isoforms Ia and Ic could also exhibit similar response to opposing mechanical load, as suggested by their similar biochemical and mechanical properties [50,51]. Interestingly, the force sensing ability of class I myosins is further tuned by other regulatory cues, which highlights the interplay between the various levels of myosin regulation. For instance, the force sensitivity of myosin Ib has been shown to be further modulated by alternative splicing within its light chain binding domain [45], as described in more detail in the previous section. Moreover, as it will be discussed in detail below, changes in Ca^2+^ concentration have been proposed to regulate the load bearing capability of myosin Ic by triggering its conversion between a rigid and a flexible conformer [52].

The ability of class I myosins to sense tension allows them to adapt to their local environment, in order perform their various intracellular roles. For instance, in intestinal brush border epithelial cells, myosin Ia has been shown to play a key role in the adhesion between the plasma membrane and underlying actin cytoskeleton, and in maintaining the membrane tension and structural stability of microvilli [53]. Interestingly, other class I myosins, including the long tail myosin Ie, have also been found able to maintain membrane tension. Moreover, myosin Ic is localised to stereocilia [54], the actin-based membrane protrusions of the inner ear hair cells that control the response to sound. There, myosin Ic functions as a tension sensitive molecular tether and is essential for the process of adaptation [51,55,56]. Finally, myosin Ic is involved in the exocytic pathway, where it has been proposed to anchor the GLUT4 exocytic vesicles between the cortical actin cytoskeleton and the plasma membrane prior to fusion [57].

Another myosin, that has shown to be regulated by load, is the minus-end directed myosin VI. Using optical trapping on myosin VI dimers, Altman et al. have showed that applying backward (plus-end directed) loads that are greater than the force required to prevent processive movement (stalling load) can significantly decrease the stepping kinetics of myosin VI, in a way that is dependent on the ATP and ADP concentration [58]. This is due to a load-induced increase in the ADP binding rate with a simultaneous decrease in the ATP binding rate. Therefore, as the load increases, the time myosin VI remains bound to actin increases. These *in vitro* observations suggest that, in the cell, when myosin VI is subjected to high load, its mode of action can be switched from a cargo transporter to a structural anchor. This model could explain many cellular and *in vivo* observations of the biological functions of this myosin. For instance, myosin VI has been shown to be essential for the normal function of the auditory and vestibular system in mammals. This role has been proposed to rely on the ability of this motor to maintain the structural integrity of stereocilia of the inner ear hair cells, by anchoring the plasma membrane at the base of stereocilia and/or stabilizing the stereocilia position on the apical region of the cell [59,60]. Although such a structural role of myosin VI has been also suggested in zebrafish [61], direct evidence is still lacking. Moreover, during spermatid individualization, the final step of *Drosophila* spermatogenesis, myosin VI has been proposed to stabilize the branched actin network of the actin cones [62]. This has been shown to be essential for individualization and male fertility [63]. As it will be described in more detail in the next section, phosphorylation of the myosin VI motor domain has been shown to enhance actin anchoring and thus, it has been proposed to be the mechanistic switch that coverts myosin VI from a cargo transporter to an actin tether [64]. Other roles of myosin VI that could reflect its membrane-to-actin anchoring function include: a) its role in the assembly of cadherin cell-to-cell contacts in epithelial cells [65], b) its function in opposing the microtubule-based transport of mitochondria along the neuronal axons by anchoring them to actin [66] and c) more recently, its ability to trigger the assembly of F-actin cages around dysfunctional mitochondria, in order to isolate them and prevent their fusion with neighbouring undamaged pools [67]. Although all these studies point towards the direction of an anchoring role for myosin VI, the evidence and the underlying mechanism remain elusive. In addition, the oligomeric state of myosin VI while performing its anchoring roles also remains unclear.

Force seems to be a regulatory cue that, in combination with other regulatory mechanisms, can tune the cellular functions of unconventional myosins. Although a lot of evidence has been gathered to date, the physiological mechanistic details of how myosins act as anchors remain to be discovered. This would require an approach that integrates *in vitro* biophysical data with cellular observations, using for instance force irresponsive tools. In addition to the myosins discussed in this section, the responsiveness to force of other unconventional myosins, traditionally established as transporters, also awaits to be addressed. For instance, a line of evidence suggests that myosin V in some cases plays the role of a dynamic tether rather than an organelle transporter [68,69,70,71].

## 4. Phosphorylation: A Versatile Regulatory Cue

Another regulatory mechanism of unconventional myosins is phosphorylation. Because of its post-translational, reversible nature, phosphorylation allows to finely tune the biochemical properties and intracellular functions of myosins in a timely and localized manner. To date, various phosphorylation sites have been mapped within either the motor domain or the tail of myosins.

More specifically, within the myosin head, a phosphorylation consensus site has been identified in the actin binding region [72]. This site was named TEDS site, given that only a threonine (T), a glutamate (E), an aspartate (D) or a serine (S) can be present at this position. Phosphorylation at the TEDS site has been shown to regulate the ATPase activity of class I myosins and to play a critical role in their intracellular localization and function in *Acanthamoeba* [73,74,75], in *Dictyostelium* [76,77], in budding [78] and fission yeast [79]. Similar to class I myosins, myosin VI contains a threonine at the TEDS site, T406 in mouse or T405 in humans. This site was shown to be phosphorylated by p21-activated kinase 3 (PAK3) *in vitro* [80]. T406 has been also proposed to be phosphorylated *in vivo* following stimulation of A431 cells with epidermal growth factor (EGF), resulting in the translocation of Myosin VI in membrane ruffles [81]. Interestingly, *in vitro* kinetic studies showed that T406 phosphorylation does not alter the actin-activated ATPase activity of Myosin VI [80,82]. However, cellular studies have suggested that the phosphorylation of this residue *in vivo* could regulate its localization, its mode of function and its interaction with the actin cytoskeleton [64,81]. Indeed, overexpression of the phosphomimic mutant T406E has been shown to induce clustering of uncoated endocytic vesicles at the cell periphery and a localized increase in F-actin density by preventing its depolymerisation [64]. PAK-mediated phosphorylation of the motor domain has been also proposed to regulate the localization of myosin VI to actin-rich, membrane ruffles, induced during Salmonella invasion [83]. There, phosphorylated myosin VI is suggested to regulate the phosphoinositide composition at the sites of invasion. This contradiction between *in vitro* and cellular studies could be attributed to the considerably more elaborate intracellular environment. Alternatively, the use of phosphomimic mutants might not accurately reflect the effect of phosphorylation, which has been already demonstrated *in vitro* [82,84].

Another myosin known to be regulated by phosphorylation of its motor domain is myosin III, a myosin localized in the photoreceptor cells of the eye [47] and the stereocilia of the inner ear hair cells [85]. Myosin III is unique in that it contains an N-terminal serine-threonine kinase domain, just before its motor domain [86]. This kinase domain has been shown to be responsible for autophosphorylation of myosin III on the actin-binding region of its motor domain [87,88,89]. This autophosphorylation has been reported to negatively regulate the ATPase activity of myosin III and its binding affinity for actin *in vitro* [89,90,91]. Consistently, in cells, phosphorylation of the motor domain has been suggested to negatively regulate the intracellular localization of myosin III on the tips of filopodia [92] or microvilli [93], as well as to modulate the density of filopodia at the cell periphery [92].

The regulation of myosins by phosphorylation is not limited within the motor domain. Various phosphorylation sites have been identified to date within the tail of many myosins. Growing evidence in the past decade suggests that tail phosphorylation is another important mechanism regulating the intracellular localization and function of myosins. For instance, the serine 1650 (S1650) in the GTD of myosin V has been reported to both positively and negatively regulate the function of myosin V. In mouse adipocytes, the insulin-stimulated phosphorylation of myosin Va at S1650 by Akt2 has been suggested to control the myosin V-mediated translocation of GLUT4 vesicles to the cell surface and the subsequent glucose uptake [94]. In contrast, in *Xenopus* melanophores, the cell cycle-dependent phosphorylation of S1650 by the Ca^2+^/calmodulin-dependent protein kinase II (CaMKII) has been suggested to negatively regulate the association of myosin V to melanosomes [95]. Phosphorylation of S1650 in myosin Va has also been shown to control the targeting of nuclear myosin V onto nuclear speckles [96]. Moreover, phosphorylation of the budding yeast class I myosin myo5 at serine 1205 (S1205) by casein kinase 2 α subunit (Cka2) negatively regulates the myo5-induced Arp2/3-dependent actin polymerization, which is critical for endocytic budding [97]. Furthermore, in myosin VI, a phosphorylation TINT motif, which includes threonine 1088 and threonine 1091, has been identified within its CBD [98]. Similar to the motor’s TEDS site, this motif has been suggested to be also phosphorylated by PAK. Experiments using phosphomimic mutants suggested that TINT phosphorylation negatively regulates the association of myosin VI with its binding partner optineurin. More recently, Tomatis and colleagues have identified a DYD phosphorylation motif, which includes the tyrosine 1114 (Y1114) and is present only in the SI isoform of myosin VI. Phosphorylation of this motif by a Src kinase family member, probably c-Src kinase, has been proposed to modulate the ability of myosin VI to anchor the secretory granules on the cortical actin network in neurosecretory cells, positively regulating neuroexocytosis [27]. Finally, more recently, myosin VI has been identified as target of the p38 MAPK pathway, downstream TNF-α stimulation in neutrophils [99]. This observation opens new opportunities in identifying and characterizing further regulatory phosphorylation sites on myosin VI.

Phosphorylation has been emerging as an important regulatory mechanism for unconventional myosins. The research conducted to date highlights how versatile this mechanism is, varying between the classes of myosin, the isoforms, their effect upon myosins’ function and the upstream stimulus. However, there is still a lot to learn about how this post-translational modification can finely tune unconventional myosins. For instance, the effect of phosphorylation on the activation, conformation and oligomeric state of many myosins remain unclear. Moreover, the mechanistic details of how phosphorylation is integrated with other regulatory mechanisms, such as isoform splicing and interactions with binding partners, also remain elusive. To address these questions would require a global approach that integrates biochemistry, quantitative cell biology, structural and biophysical studies.

## 5. Regulation Within the Local Environment: Divalent Cations

Another way to regulate the function of a myosin is through the temporal changes of its local environment, which includes the concentration of divalent cations.

Mg^2+^ concentration has been shown to affect the biochemical properties of various myosins *in vitro* [9]. Increased concentration of free Mg^2+^ has been shown to result in a decreased actin-activated ATPase activity of the high duty ratio myosins, like myosin Va [100,101], myosin VI [102], myosin VII [103]. This is due to the Mg^2+^-induced increase in the affinity for ADP. High Mg^2+^ levels have been associated with decreased velocity of myosin V [100,101] and also with conformational changes within myosin V active site [104]. Similarly, free Mg^2+^ has been shown to regulate the motile activity of myosins Id and Ie by inhibiting ADP release [76,77]. Interestingly, the concentration of free Mg^2+^ has been proposed to modulate the effect of TEDS phosphorylation on motor activity, by enhancing the tension bearing ability of the motor and suppressing its fast motor activity [76,77]. Although these observations suggest a potent role of Mg^2+^ as a regulator of unconventional myosins, they are only based on *in vitro* experimentation. Therefore, expanding this knowledge in the context of the intracellular environment would be key in fully addressing the role of Mg^2+^ in myosin regulation.

Another second messenger known to regulate the function of unconventional myosins is Ca^2+^. Ca^2+^ affects the biochemical properties of myosins, mainly through the interaction with CaM or CaM-like light chains, which are associated with the IQ motifs on the myosins’ neck region. Most of these light chains consists of four EF-hand motifs that have been shown to bind Ca^2+^ and/or Mg^2+^ [9]. Mainly based on *in vitro* studies, Ca^2+^ has been reported to regulate the unconventional myosins in various ways.

First, Ca^2+^ has been proposed to have a direct effect on the activity of the motor domain of myosin Ic [105] and myosin Ib [106,107], by decreasing the affinity for ADP. More specifically, myosin Ic was shown to have a calcium-sensitive ATP hydrolysis step, whereby Ca^2+^ inhibits the rate of the ATP hydrolysis and increases the rate of ADP release [105]. This effect was suggested to be due to Ca^2+^ binding to the CaM associated with the fist IQ domain of the neck. This was further supported by the crystal structure of the human myosin Ic motor in complex with CaM, which revealed a direct contact between the motor and CaM [108]. Such direct interaction between motor and the CaM bound on the first IQ motif has been also reported for myosin Ib [109]. This direct effect of Ca^2+^ on myosin Ic motor activity has been proposed to allow this myosin to play its key role in the adaptation response of the inner ear.

In addition to this direct effect, Ca^2+^ regulates the function of myosins by modulating the association of CaM with the neck region. This interaction stabilizes the α-helical IQ regions of the neck, providing the required mechanical stiffness. This allows the neck to act as a lever arm that transduces force from the motor domain to the tail. Ca^2+^-binding to CaM induces a conformational change within its EF hand motifs, which in turn leads to either the dissociation of CaM or, in some cases, to the strengthening of the interaction [11]. In this way, Ca^2+^ has been proposed to regulate the ability of myosins to produce movement and sense tension. This “clutch model” has first been proposed for the load sensitive class I myosins [110]. Indeed, *in vitro* studies have shown that physiological Ca^2+^ concentrations induce dissociation of at least one CaM from myosin Ia and Ic and lead to decreased motility [9]. This model has been further supported by structural studies by Lu and colleagues on myosin Ic. Their study has revealed that a post IQ3 region, situated just before the c-terminal PH domain, binds CaM together with IQ3 [52]. They have proposed a model whereby Ca^2+^ modulates the conformation of myosin Ic in order to switch between a large load/high duty ratio and a small load/low duty ratio mode of action: at low Ca^2+^ levels, the binding of 3 CaMs onto its neck region provide a rigid extended conformation suitable for transmitting force between the plasma membrane and the cortical actin network, tethering vesicles or sensing tension in stereocilia and microvilli [6]. Increase in Ca^2+^ levels would induce conformational rearrangements and possibly dissociation of some of the CaMs, resulting in a more flexible conformation which has lower load bearing capability.

Ca^2+^ has been also proposed to regulate myosins by relieving auto-inhibition. For instance, Ca^2+^ has been shown to activate the ATPase activity of Drosophila myosin VIIa by inducing a conformational change to the CaM bound to the distal IQ motif and in this way abolishing the inhibitory interaction between the head and the tail [23]. More recently, structural studies on the single α helix (SAH) of myosin VIIa revealed that Ca^2+^-binding to the IQ5-bound CaM softens the conformation of the IQ5-SAH, providing increased flexibility of the lever arm [111]. Micromolar Ca^2+^ concentrations have been shown to stimulate the actin-activated ATPase activity of myosin Va [21,112]. This is achieved by releasing the auto-inhibitory interaction between its motor and its globular tail domain (GTD) [113,114]. This effect was attributed to Ca^2+^ binding to the CaM associated with the first IQ motif, which together act as a Ca^2+^ sensor [115,116]. Similarly, Ca^2+^-induced relief of auto-inhibition and the resulting ATPase activation and association with cargo has been observed for myosin Vb *in vivo* and *in vitro* [117]. Interestingly, although Ca^2+^ does not directly affect the activity of the motor of myosin V [21,118], it significantly impairs its processivity *in vitro* by inducing dissociation of CaM from one or more of the IQ motifs, and therefore compromising the stiffness of the neck [118,119,120]. Therefore, although Ca^2+^ activates myosin V through release of the auto-inhibition, it also results in a mechanically compromised myosin. This raises questions about the physiological relevance of Ca^2+^ as an activator of myosin V, although the *in vivo* data on myosin Vb [117] point to this direction. Finally, myosin VI is also affected by Ca^2+^. For instance, its ability to bind lipids has been found to be Ca^2+^-dependent [32]. Moreover, Ca^2+^ has been shown to negatively regulate the actin-translocating activity of myosin VI [80], to reduce the ADP release rate and to impede the coordination between the two heads of the myosin VI dimer [82]. More recently, Ca^2+^ has been proposed to regulate the conformation of myosin VI [121]. Batters et al. proposed a model whereby Ca^2+^ binding to CaM triggers a conformation change which converts myosin VI from the auto-inhibited compact state to a state primed for cargo binding, which however is mechanically inactive. This model though has been contradicted by Fili et al., who observed that Ca^2+^ has no effect on the conversion of myosin VI from its back-folded to its open conformation, when the same assays were performed [14].

In summary, although Ca^2+^ seems to have a significant role in the regulation of unconventional myosins *in vitro*, its regulatory mechanism *in vivo* remains to be established. Improved biosensors and tools for the localized manipulation of Ca^2+^ inside cells would be critical for assessing the physiological effect of this regulator on the conformation and activity of myosins. Finally, although CaM has been the main light chain used for studying the regulation of myosins, it is still not clear which light chain is the physiological subunit for each unconventional myosin *in vivo* [11]. Given the differential response of different light chains to Ca^2+^ concentrations, the physiological mechanism of regulation by Ca^2+^ remains to be addressed.

## 6. Regulation Within the Local Environment: Membranes and Phospholipids

Another local regulator of unconventional myosins are the phopspholipids, and in particular the phopshoinositides (PIs), which act as second messengers and targeting cues onto intracellular membranes. Myosins interact with these lipids through pleckstrin homology (PH) domains, like in the case of myosin I [122,123] and myosin X [124], or through a stretch of basic residues, in the case of myosin VI [32], located within their tail. Various studies have demonstrated how the interaction with lipids regulates the intracellular localization of myosins and, in some cases, their conformation and activity.

One example of an unconventional myosin which is regulated by lipids is myosin X, best known for its role in filopodia formation and its fast, long range intra-filopodia cargo transport activity [125]. The globular tail of myosin X contains three consecutive PH domains, which are followed by a myosin tail homology 4 (MyTH4) domain and a FERM domain [126]. While the first (PH1) and third (PH3) PH domains bind non-specifically to negatively charged phospholipids, the second one (PH2) is highly specific for phosphatidylinositol-3,4,5-triphosphate (PtdIns(3,4,5)P_3_), with the PH123 tandem binding cooperatively with high affinity to PtdIns(3,4,5)P_3_-rich membranes [124]. Interestingly, myosin X has been shown to be monomeric in solution and to adopt a back-folded auto-inhibited conformation, in which its motor domain directly interacts with the PH-FERM domain of its tail [25]. Umeki et al. elegantly demonstrated that association of the PH domain with PtdIns(3,4,5)P_3_ relieves this tail-induced auto-inhibition and activates the actin-activated ATPase activity of myosin X. This PI-induced activation has been shown to power the motility of myosin X *in vitro*, to enable its dimerization and, in cells, to allow its translocation at the filopodial tips.

Moreover, myosin VI has been shown to specifically interact with high affinity with phosphatidylinositol-4,5-diphosphate (PtdIns(4,5)P_2_) through a stretch of alternating basic (R/K) and hydrophobic residues, which is located within its CBD [32]. This interaction, which has been shown to be Ca^2+^ -dependent, has been found to be important for the recruitment of the protein in clathrin coated structures. In addition to regulating the intracellular localization of myosin VI, PtdIns(4,5)P_2_ binding has been also shown to induce 30% increase in α-helicity, as well as dimerization of the tail, as demonstrated by liposome binding assays [32]. Moreover, another group has shown that exposure of myosin VI to lipids can induce conformational changes within the coiled-coil region which follows the IQ motifs, namely the lever arm extension (LAE): the reversible expansion of the LAE from a compact helix bundle to a partially extended rod-like structure [127]. This lipid-induced conformational change has been suggested to provide myosin VI with the flexibility to undertake large and variable steps of 30–36 nm. Although these observations highlight the role of lipids in the regulation of myosin VI, the mechanistic details of how this regulation occurs *in vivo* still remain to be addressed.

Class I myosins have been also shown to interact with membranes, although the binding mechanism is highly diverse and isoform specific. In myosin Ic, membrane association is mediated by a PtdIns(4,5)P_2_ specific PH-like domain [123] and basic residues in other regions of the tail which form non-specific electrostatic interactions with anionic phospholipids [128]. PH-like domains have been identified in both short and long tailed myosin I isoforms [129,130,131]. However, the lipid specificities of these domains and their role in membrane binding highly depends on the isoform. For instance, in myosin Ig, the PH-like domain alone is not sufficient for membrane localization, as it also requires two flanking regions, N-terminal and C-terminal of the domain [131]. In contrast, in the long-tailed myosin Ie, which tightly binds not only PIs but phosphatidylserine as well, the PH domain signature residues do not seem to be critical for membrane association [122]. Instead, membrane association of myosin Ie has been proposed to rely on high-affinity electrostatic interactions. Although, the interaction of class I myosin with lipids has been extensively studied, there are still a lot to be discovered in relation to how this interaction regulates the intracellular function of these proteins. For instance, it has been recently proposed that the interaction of myosin Ic with PIs is indispensable for its import into the cell nucleus [132]. In addition to controlling intracellular localization, lipid binding has also been shown to affect the mechanical properties of myosin Ic *in vitro*. Binding of myosin Ic to fluid lipid bilayers containing PtdIns(4,5)P_2_ has been shown to enhance the asymmetric, counter clockwise motility of actin [133]. Interestingly, the actin gliding velocity was found to decrease with increasing levels of PtdIns(4,5)P_2_, suggesting a dependence on membrane affinity. These observations highlight how membrane association can provide myosins with the freedom to adopt conformations that enable an asymmetric working stroke, which could be required for some intracellular roles.

Although a lot has been learnt about how lipids can act as regulatory cues for some unconventional myosins, it is quite clear that the mode of regulation varies depending on the type of myosin. Therefore, there is still a lot to discover about how membranes, their composition and curvature, can modulate the conformation, oligomeric state and activity of other unconventional myosins and how this regulation correlates their intracellular function.

## 7. Tail Dependent Regulation Through the Interaction with Binding Partners

In addition to the above-mentioned regulatory mechanisms, a significant part of the regulation of unconventional myosins relies on the interaction of their tail with adaptor proteins. Each myosin can interact with a diverse range of binding partners, each being the specific link of the myosin to a defined cargo. The interaction with these adaptor proteins, which are under tight spatio-temporal regulation, modulate myosins in a wide range of ways: by triggering conformational changes, activating motor activity, inducing dimerization, and defining intracellular localization or cargo. In this section, we will be summarizing the current knowledge on the diverse and versatile mechanisms through which binding partners can regulate the function of myosins, as demonstrated by some characteristic examples.

### 7.1. Regulation by Binding Partners: Myosin V

The dimeric myosin V, which in solution adopts a back-folded, auto-inhibitory conformation with low actin-activated ATPase activity [21,22], is a well-established example of a myosin under such regulation. As described above, Ca^2+^ has been shown to activate myosin Va *in vitro* [21,112], by releasing this auto-inhibition [113,114]. However, the *in vivo* relevance of such activation is still debated [117,118,119,120]. Instead, the interaction of myosin V with its cargo adaptors has been accepted as the main physiological mechanism of activation. Binding partners are proposed to destabilize the motor–GTD interaction, making the GTD interaction sites accessible for binding. Similar effect has been suggested to occur through the *in vitro* surface immobilization of myosin V, which has been observed to activate myosin V [21]. Within its GTD, myosin V contains two helical-bundled binding sites, namely subdomain 1 (SD-1) and subdomain 2 (SD-2), which mediate its interaction with a broad range of cargo adaptors [134,135]. Unique structural features within these domains, some conserved through evolution and others isoform specific, enable a highly specific cargo recognition mechanism. Melanophilin, for instance, specifically binds to myosin Va isoform and stimulates its actin-activated ATPase activity [40]. Through its interaction with the small GTPase Rab27a, melanophilin serves as a partner that docks myosin Va onto melanosomes [136,137]. Given that the binding site to melanophilin (within SD-1) [134] is distinct from the interaction site with the motor domain (within SD-2) [114], it has been proposed that melanophilin relieves the auto-inhibitory motor-tail interaction by inducing allosteric conformational changes [138]. Interestingly, binding to melanophilin requires the presence of the melanocyte-specific alternatively spliced exon F in the tail myosin-Va [39,139], highlighting how isoform splicing regulates binding partner specificity. Apart from activating the motor activity, melanophilin has been also shown to alter the biophysical properties of the motor, by decreasing its speed of movement, increasing its run length and enhancing its processivity by providing additional anchoring onto actin [140]. Within the SD-1, but on a distinct site and through a different interacting mechanism, myosin Va also interacts with isoform-specific adaptor Rab interacting lysosomal protein-like 2 (RILPL2) [135], which in turn binds to Rab36 [141]. Myosin V also directly interacts with Rab family of small GTPases, including Rab11 [134], Rab8 [37,142,143] and Rab10 (Roland JT, J Biol Chem, 2009). The interaction with Rab11 has been the most extensively studied and has been shown to occur through the SD-2. Although the binding site is conserved though evolution including the yeast homolog Myo2p, it is limited to isoforms Va and Vb, but not myosin Vc [37]. Structural studies have revealed that myosin Vb binds both the GDP- and GTP- forms of Rab11, with around 30-fold higher affinity for the GTP-form [134]. Although the binding to Rab11 has been proposed to activate the Drosophila myosin V (DmM5) [144], the mechanism of this activation and the effect on the motor properties remains to be addressed. Similarly, given the broad range of binding partners and the diverse mechanisms underlying their interaction with the myosin V isoforms, it remains unclear whether the melanophilin-induced activation and the changes in myosin Va’s biophysical properties is a generally applied mechanism or not.

In addition to their function as targeting cues, cargo adaptors regulate myosins by increasing their local concentration, which in turn allows either their direct dimerization or their conversion into processive dimeric complexes that are able to transport their cargo. The latter case is exemplified by the intrinsically monomeric, non-processive *Saccharomyces Cerevisiae* class V myosin, Myo4p [145], which transports mRNAs along actin cables to the tip of the bud [146]. Myo4p association with its cargo is unique in that, it first forms a single-headed motor complex with its partner She3p, which prevents its self-dimerization [147], and then it assembles into a processive two single-headed Myo4p–She3p motor complex through the tetrameric mRNA-binding protein She2p [148]. *In vitro* reconstitution experiments elegantly demonstrated that the mRNA cargo itself contributes to the stability of the complex, with the run length and frequency increasing with the number of motors recruited [149]. Although this cargo association has been shown in yeast, it might be a mechanism followed by other monomeric myosins in higher eukaryotes.

### 7.2. Regulation by Binding Partners: Myosin VII

Activation by binding partners has been also proposed for myosin VII. Despite the presence of short predicted coiled-coil region preceding its globular tail domain, structural, biochemical and cellular studies have shown that myosin VII exists in solution as an auto-inhibited, back-folded monomer [17,23,24]. Apart from the previously mentioned Ca^2+^ induced release of auto-inhibition, cargo adaptors have been also suggested to activate the cargo-transporter function of myosin VII. One such adaptor is MyRIP (myosin-VIIa- and Rab-interacting protein), which in turn binds to Rab27. This ternary complex has been proposed to link the retinal melanocytes to the actin cytoskeleton and to mediate their intracellular trafficking in retinal pigment epithelium cells [150]. The association of myosin VIIa with MyRIP has been shown to induce translocation of the myosinVIIa/MyRIP/Rab27 complex to the filopodial tips [17]. Interestingly, this translocation required dimerization of myosin VIIa and its MyRIP-induced membrane association was shown to promote dimerization of myosin VIIa tails. Although, there is not yet a direct proof for cargo-induced dimerization, these observations could be explained by the increased local concentration of myosin VII monomers on the cargo. This localized clustering of monomers, which can move laterally on the fluid cargo membrane, could activate cargo motility in two ways: either by inducing dimer formation between monomers which are close enough to each other to interact or by the coordinated activity of adequate number of myosin VII monomers. The later has been already proposed for myosin VI, as it will be discussed in more detail below [151]. The mechanistic details of how binding partners regulate the oligomeric state and transport activity of myosin VII remains to be deciphered.

### 7.3. Regulation by Binding Partners: Myosin VI

Finally, another well-studied example of regulation by binding partners is myosin VI. Myosin VI is known to interact with a wide range of binding partners that mediate its broad spectrum of cellular functions, including endocytosis, exocytosis, cell motility, organelle morphology and transcription [14,152,153,154]. Apart from defining the subcellular localization of myosin VI, these interactions have been now shown to also regulate its conformation, oligomeric state and function. Similar to myosin VII, myosin VI has been shown *in vitro* to be monomeric in solution, [155] and proposed to adopt a back-folded conformation [16,121,155,156]. More recently, biochemical analysis confirmed that indeed the tail of myosin VI is able to back-fold, whereby the CBD forms intramolecular contacts with the N-terminal part of the tail as well as the motor domain [14]. Most importantly, the back-folded form of myosin VI has been shown to be a physiological conformation of the protein, occurring inside cells [14]. Although widely proposed, the role of binding partners in the regulation of this back-folded conformation has remained speculative until recently. Fili et al. has demonstrated that the binding partner nuclear dot protein 52 (NDP52) [157], can trigger the unfolding of myosin VI by destabilizing the interaction between the CBD and the rest of the protein [14] (Figure 3). This conformational change exposes the DNA binding sites within the CBD, which are required for the role of myosin VI in transcription.

Apart from exposing additional binding sites that are occluded by the intramolecular back-folding, myosin VI unfolding has been proposed to expose dimerization sites, hence enabling dimer formation. Similar to myosin VII and X, myosin VI contains a predicted stable single α helix (SAH) [18], the proximal end of which has been proposed to mediate its dimerization [15]. The ability of full length myosin VI to form stable, processive dimers has been demonstrated *in vitro* following actin-induced monomer clustering [158] or suggested by the *in vitro* dimerization of tails on lipid vesicles [32] and the close proximity of CBD constructs on uncoated vesicles (UCV) *in vivo* [159]. However, the first evidence of partner induced dimerization was demonstrated by the structural studies of Yu et al. They demonstrated that the interaction of the CBD with a fragment of the clathrin-coated vesicle adaptor Disabled- 2 (Dab2) results in the formation of a symmetric 2:2 complex: the 2 CBDs being held together by the two Dab2 fragments with little interaction between the CBD themselves [160]. Various models [15,16,108] have been proposed to describe how dimerization results is a processive dimer able to take 30–36 nm steps on actin [161,162]. The latest suggests that dimerization induces unfolding of the three-helix bundle of the lever arm extension (LAE) [15]. Recently, biochemical analysis has revealed the mechanistic details of partner-induced dimerization of myosin VI: upon release of the back-folded conformation, NDP52 enables dimerization of two myosin VI monomers by exposing the dimerization site, which is otherwise masked in the back-folded conformation [14] (Figure 3). This dimerization in turn induces the mechanical activation of the motor. The data suggest that this dimerization could occur in different formats, with two monomers dimerized around a single partner or dimerization between two monomer-partner complexes. Interestingly, the ability of binding partners to trigger dimerization of myosin VI is not isoform specific: Dab2 has been shown to induce dimerization of both non-insert (NI) and large insert (LI) isoform of myosin VI [31]. Finally, novel insights into the regulation of myosin VI were recently gained when it was shown that the two hotspots of interaction with its binding partners, namely the RRL and the WWY motifs [32,157,163], exhibit different affinities for their partners, with the RRL being the high affinity binding site [31]. This differential affinity provides a mechanism for binding partner selectivity for the NI isoform (Figure 3). Unlike the LI isoform, in which, as described above, the high affinity RRL site is masked by the LI [30], this isoform has both motifs equally accessible for binding. In this way, this regulatory mechanism favors the high affinity RRL binding partners for unfolding and dimerizing the NI isoform. In addition, it allows the competition between binding partners to be driven by the local partner concentration. This can impact on myosin VI function in health and disease, as it has been proposed to be case in cancer, where the loss of Dab2 enhances myosin VI–mediated transcription [31].

To date, a lot of insight has been gained on how binding partners can regulate unconventional myosins. It has becoming increasingly clear that this regulation has evolved to occur in many different ways, which also vary depending on the cargo adaptor and the myosin. In addition, the interplay between binding partner regulation and other regulatory pathways, such as isoform splicing, highlight the extent of complexity and suggests that there is still a lot to learn about this finely tuned regulatory mechanism.

## 8. Conclusion and Perspectives

Unconventional myosins are multi-functional proteins and therefore, they need to be under tight regulation. As summarized in Figure 2 and Table 1, a broad range of regulatory mechanisms have evolved for this purpose. During the past two decades, biophysical, biochemical, structural and cellular observations have provided insights into the mechanistic details of how these modes of regulation are applied to specific classes of myosins (Table 1). Although many aspects of this regulation are myosin specific, one could recognize common regulatory patterns across the different classes: for instance, the activation through auto-inhibition release, the effect of isoform splicing, Ca^2+^ levels and phosphorylation. Moreover, these regulatory modes are not restricted to unconventional myosins, but also modulate the conventional smooth and non-muscle myosin II. As reviewed in detail in Heissler and Sellers [9], myosin II adopts a compact auto-inhibited conformation, which is relieved through phosphorylation of its regulatory light chain (RLC), powering its ATPase activity and triggering filament formation. In addition to RLC phosphorylation, the function of non-muscle myosin II is also regulated through alternative splicing within its motor domain, as well as though binding partner interactions, like with the Ca^2+^ binding protein S1004A. However, despite the recent advances in our knowledge, many aspects of how these common regulatory patterns are finely tuned to specifically serve the properties and intracellular functions of each myosin remain unknown. Many of the currently proposed models of regulation are based on either solely *in vitro* biochemical/biophysical studies or on solely cellular observations. In order to acquire a rounded understanding of how a myosin is regulated, the mechanistic details that have been deciphered *in vitro* would need to be translated into a physiological mechanism *in vivo*. Similarly, models built on *in vivo* observations would need to be enriched with mechanistic details obtained *in vitro* though biophysical approaches.

Furthermore, given the key role that unconventional myosins play in vital intracellular processes, it is not a surprise that their malfunction and dysregulation is associated with pathological phenotypes. Dissecting the regulatory mechanism that control these motor proteins not only enhances our understanding on how they function, but also allow us to establish their role in disease. The striking differences in the properties, intracellular localizations and functions of splice variants of unconventional myosins highlight how dysfunction in alternative splicing can lead to disease. One example is myosin VI, the NI isoform of which is the only isoform overexpressed in aggressive, metastatic cancers [30]. This has been recently attributed to the role of this isoform in transcription [14]. Further research on the link of the splice variants of other unconventional myosins with pathological conditions, would shed more light into how this regulatory mechanism can lead to disease and potentially how this can be prevented.

Finally, it has become increasingly evident that the various regulatory mechanisms do not act in isolation. Instead, there is significant interplay between them, as it has been highlighted across this review (Table 2). This interplay aims to achieve additional levels of regulation, which is required for such multi-functional proteins. Although we have recently started to acquire understanding of how these regulatory paths converge in the case of some myosins, the extent of this interplay across the whole family and the underlying mechanistic details remain elusive. Dissecting the interplay between the various regulatory modes will, not only enhance our knowledge on myosin regulation, but would also pave the way in finding new and more effective therapeutic routes to tackle disease.

## Figures and Tables

**Figure 1 ijms-21-00067-f001:**
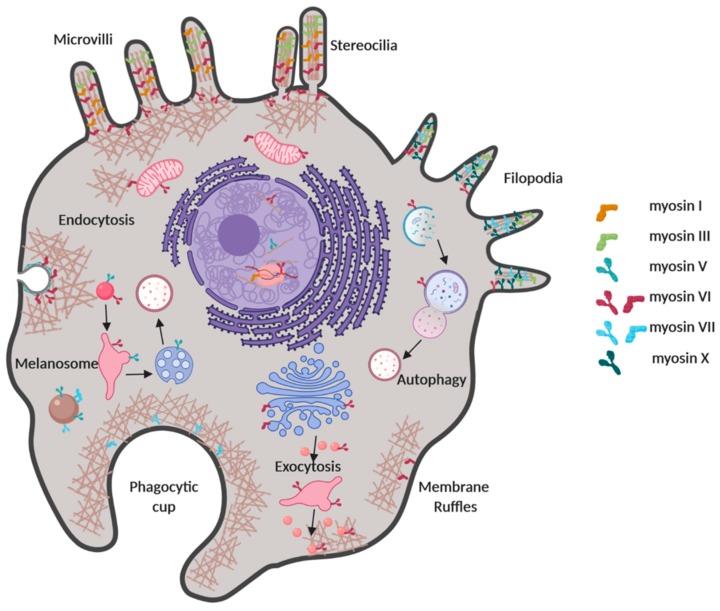
Schematic representation of the multiple roles that unconventional myosins fulfil in key cellular processes. The myosins, whose regulation is discussed in this review, are illustrated as examples. Depending on their structural and mechano-enzymatic features, each myosin can function as a monomer or dimer and can act as a cargo transporter, molecular anchor, actin cross-linker or tension sensor. Note that each myosin can be assigned to a broad range of intracellular roles. Because of this multi-functionality, myosins require tight spatial and temporal regulation, which is achieved at multiple levels.

**Figure 2 ijms-21-00067-f002:**
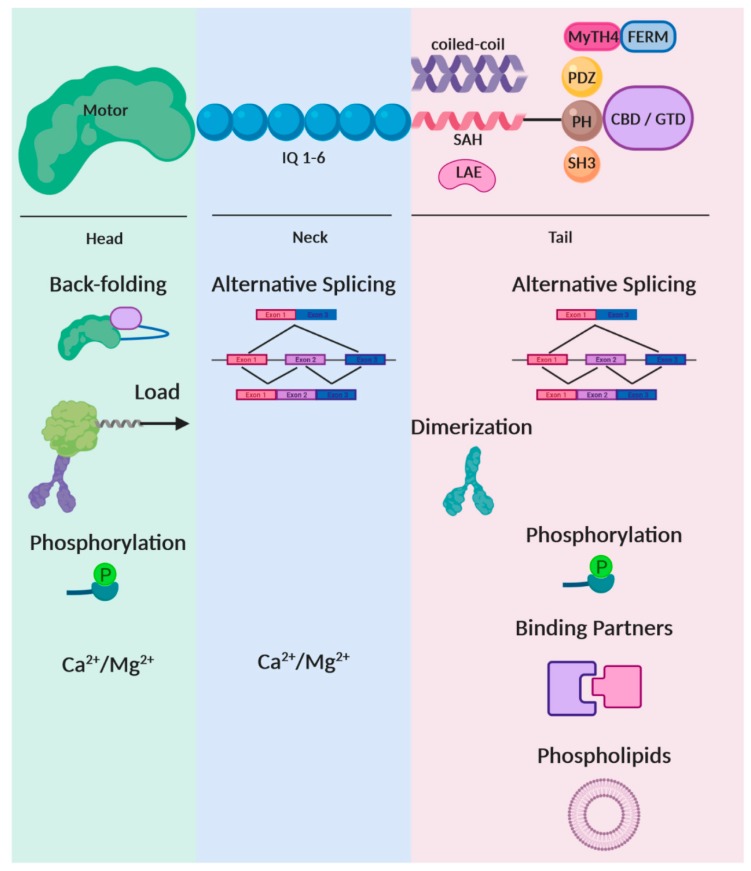
Schematic representation of the architecture of unconventional myosins and their various mechanisms of regulation. Myosins consist of three main regions: the highly conserved motor head domain; the neck domain which can include one to six IQ motifs, which bind calmodulin or other light chains; and the highly diverse tail domain which can include coiled-coil regions that mediate dimerization, helical lever arm extension (LAE) regions, single α-helix (SAH) regions that can also serve as lever arm extensions, and various lipid- and/or protein-binding domains that mediate the interaction of myosins with membranes and their cargo adaptors, such as pleckstrin homology (PH), Src homology 3 (SH3) domain, myosin tail homology 4–band 4.1, ezrin, radixin, moesin (MyTH4-FERM) domain, PSD95/Dlg/ZO1 (PDZ) domain, followed by a cargo binding domains (CBD) in myosin VI or globular tail domain (GTD) in myosin V. Various regulatory mechanisms are deployed in order to tune the activity, conformation, oligomeric state, intracellular localization and function of myosins. These mechanisms are depicted here in relation to the region of the myosin which gets regulated.

**Figure 3 ijms-21-00067-f003:**
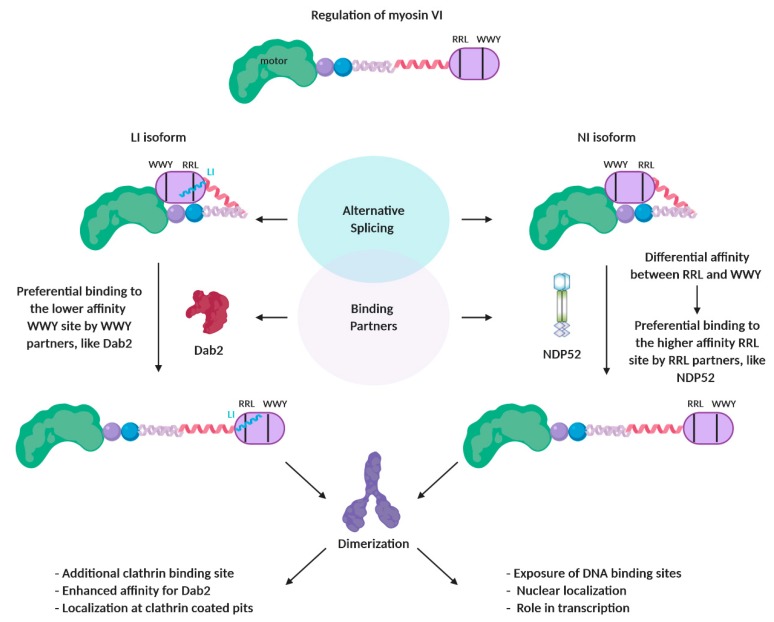
Regulation of myosin VI through the interplay between alternative splicing and binding partner interaction. Alternative splicing of myosin VI results in four different isoforms, including a non-insert (NI) and a large insert (LI) isoform. In the NI isoform, the two 3 amino acid partner binding motifs RRL and WWY are equally accessible. In contrast, in the LI isoform, the LI helix masks the RRL motif. The motifs are characterized by different affinities, with the RRL motif being the high affinity site. In the case of the LI isoform, alternative splicing interferes with binding partner selectivity in two ways: a) by masking the high affinity RRL site, therefore favoring interactions with the lower affinity WWY partners, like Dab2 and b) by directly enhancing the interaction with Dab2. In the case of the NI isoform, partner selectivity is dictated by the differential affinity between the two sites, hence favoring the interactions with RRL partners like NDP52. This also allows the competition between binding partners to be driven in the cell by the localized concentration of binding partners. The interaction with the selected partner triggers the conversion of each isoform from a folded to an unfolded conformation, which is capable of dimerization. In addition, this differential binding partner selectivity dictates to each myosin VI isoform a different intracellular localization and function. Illustrated model as proposed by Fili et al., 2017 [14] and Fili et al., 2019 [31].

**Table 1 ijms-21-00067-t001:** Modes of regulation of unconventional myosins. This table summarises the various modes of regulation of unconventional myosins, as discussed in this review. On the basis of the current literature, these are presented along with the specific myosin property (enzymatic, mechanical, structural and/or biological) that is regulated through these mechanisms.

Myosin	Mode of Regulation	Properties
**Class I**	Alternative splicing	Motility Tension sensing
	Applied load	Duty ratio Tension sensing
	Motor domain phosphorylation	ATPase activity Intracellular localization and function
	Phosphorylation within the tail	Arp2/3-dependent actin polymerization
	Mg^2+^	Motility (myosin Id, Ie)
	Ca^2+^	ATPase cycle (myosin Ic, Ib) Stiffness of the neck–duty ratio Ability to produce movement, to anchor and to sense tension (myosin Ia, Ic)
	Phospholipids	Intracellular localization (myosin Ic) Actin motility (myosin Ic)
**Class III**	Alternative splicing within the neck and tail region (myosin IIIb)	Number of IQ motif and tail length
	Motor domain phosphorylation	ATPase activity Affinity to actin Intracellular localization and function
**Class V**	Alternative splicing in the region between the neck and the GTD	Binding partner/cargo selectivity
	Phosphorylation within the tail	Intracellular localization
	Mg^2+^	ATPase activity, velocity, conformation
	Ca^2+^	ATPase activity; Auto-inhibition relief (myosin Va, Vb) Stiffness of the neck
	Binding partners	ATPase activity; Auto-inhibition relief Velocity, run length, processivity (myosin Va by melanophilin) Assembly of monomers into processive dimers (Myo4p)
**Class VI**	Alternative splicing within the CBD	Intracellular localization Binding partner selectivity
	Applied load	Duty ratio Switch from a cargo transporter to an anchor
	Motor domain phosphorylation	Intracellular localization Interaction with actin
	Phosphorylation within CBD	Interaction with binding partners Intracellular localization
	Mg^2+^	ATPase activity
	Ca^2+^	Actin-translocating activity ATPase cycle Coordination between heads in the dimer (gating)
	Phospholipids (PtdIns(4,5)P_2_	Structural changes: increase in α-helicity Dimerization Conformation changes within LAE; step size Intracellular localization
	Binding partners	Conformation change: from a back-folded to an open conformation Exposure of additional sites within CBD, masked in the back-folder conformation: DNA binding sites, dimerization site Dimerization Binding partner selectivity: though differential affinities Intracellular localization
**Class VII**	Mg^2+^	ATPase activity
	Ca^2+^	ATPase activity; Auto-inhibition relief Flexibility of the lever arm
	Binding partners	ATPase activity; Auto-inhibition relief Intracellular localization
**Class X**	Phospholipids (PtdIns(3,4,5)P_3_)	Auto-inhibition relief Motility; dimerization Intracellular localization
**Class XV**	Alternative splicing N-terminal to the motor domain (myosin XVa)	None defined
**Class XVIII**	Alternative splicing within the tail	Intracellular localization

**Table 2 ijms-21-00067-t002:** Interplay between various modes of regulation. The regulatory mechanisms that modulate the properties and function of unconventional myosins do not act in isolation. Instead, they converge in order to achieve a tighter and more diverse control of myosins. This table summarises examples of the interplay between different regulatory modes, along with the specific myosin function that is modulated though this interplay.

Myosin	Interplay between Modes of Regulation: Effect on Myosin Function
**Class I**	Force sensing: Alternative splicing within the neck region and applied load (myosin Ib) Ca^2+^ concentration and applied load (myosin Ic)
**Class V**	Association with cargo: Alternative splicing within the tail and binding partner interactions
**Class VI**	Intracellular localization and function: Alternative splicing within the CBD and phosphorylation (SI isoform) Alternative splicing within the CBD and binding partner interactions Anchoring: Applied load and phosphorylation within the motor domain

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
