# Peer review of "Unconventional Myosins: How Regulation Meets Function"

_ijms, 2019, doi:10.3390/ijms21010067_

Round 1
Reviewer 1 Report
This is a very up to date and comprehensive review of the regulatory strategies on non-muscle myosin. It could be improved in several ways.
1 Figure 1, the cell interior should be light grey or even white rather than lilac so that the myosin locations can be clearly seen when printed. In the current figure myosin is almost invisible printed in black and white
2 The manuscript is very much written word only; for ease of reference there should be tables for each section 2-7 summarising the myosin types and their mode of regulation referred to in the text.
3 In the final discussion the authors should draw parallels with myosin II regulation in smooth and striated muscle, that parallels many of the mechanisms found in the unconventional myosin - back folding, alternative splicing, phosphorylation, Ca2+-regulation.
Author Response
We thank the reviewer for the positive evaluation of the manuscript and for their suggestions below.
1 Figure 1, the cell interior should be light grey or even white rather than lilac so that the myosin locations can be clearly seen when printed. In the current figure myosin is almost invisible printed in black and white
>>We understand this criticism and we have made this change to Figure 1.
2 The manuscript is very much written word only; for ease of reference there should be tables for each section 2-7 summarising the myosin types and their mode of regulation referred to in the text.
>>We welcome this suggestion and we have included two tables within the manuscript.
3 In the final discussion the authors should draw parallels with myosin II regulation in smooth and striated muscle, that parallels many of the mechanisms found in the unconventional myosin - back folding, alternative splicing, phosphorylation, Ca2+-regulation.
>>A section discussing this point has been included in the final section of the review.
Reviewer 2 Report
A review by N. Fili and C.P. Toseland focuses on unconventional myosin, which plays an important role in fundamental cellular processes. Based on extensive literature, the authors examine in detail the structural organization of this myosin, the mechanisms of regulation of its function and its role as cargo transporter along the actin cytoskeleton, molecular anchor and tension sensor. The authors dwell in detail on the structural organization of the unconventional myosin and on the mechanisms of regulation of its function in the cell. In particular, the regulation of myosin function through splicing of various isoforms, interaction with its binding partners, phosphorylation, applied load and the composition of its local environment, such as ions and lipids, is analyzed. The authors demonstrate a deep understanding of the complexity of the molecular mechanisms of regulation of the functional properties of unconventional myosin in the cell.
Author Response
We thank the reviewer for the positive evaluation of our manuscript.